

# Comprehensive evaluation for the sustainable development of fresh agricultural products logistics enterprises based on combination empowerment-TOPSIS method

Dechao Sun, Xuefang Hu and Bangquan Liu

College of Digital Technology and Engineering, Ningbo University of Finance & Economics, Ningbo, Zhejiang Province, China

## ABSTRACT

To solve the problems of environmental pollution and resource waste caused by the rapid development of cold chain logistics of fresh agricultural products and improve the competitiveness of logistics enterprises in the market, a performance evaluation method of cold chain logistics enterprises based on the combined empowerment-TOPSIS was proposed. Firstly, from the five dimensions of cold supply chain capacity, service quality, economic efficiency, informatization degree and development ability, a comprehensive evaluation system of logistics enterprises' sustainable development is constructed, which consists of 16 indicators, such as storage and preservation capacity, distribution accuracy, and equipment input rate. Then, G1 method and entropy weight method are used to calculate the subjective and objective weights of the evaluation indicators, and the combined weights are calculated with the objective of minimizing the deviation of the subjective and objective weighted attributes. Finally, the TOPSIS method is used to calculate the comprehensive evaluation indicators. The results show that the established performance evaluation model can effectively evaluate the performance of fresh agricultural products logistics enterprises and provide theoretical basis for enterprise logistics management.

## INTRODUCTION

With improvements in living standards, fresh agricultural products have become increasingly popular among consumers. However, the rapid development of agricultural product logistics has created several challenges. According to recent data from the General Office of the State Council, only 1% of more than 4,000 fresh food e-commerce businesses in China are profitable, with 7% experiencing huge losses, 88% slightly negative, and 4% flat. The lack of standardized cold chain logistics and high logistics management costs are the main reasons for this trend. In China, the direct loss caused by the decay of fresh agricultural products is 100 billion yuan annually, and output waste exceeds

Corresponding author
Bangquan Liu, walle.cn@126.com

100 million tons. Given the perishable nature of fresh agricultural products, their short shelf life, and their high logistics distribution costs, it is necessary to ensure that they are transported at low temperatures during distribution. However, the implementation of this process faces several challenges, including high cold chain technology costs, difficulties in promotion, high facility requirements, coordination difficulties among cold chain logistics development subjects, and imperfect laws and standards (*McCullen, 2017*). Therefore, there is an urgent need to establish an evaluation index system suitable for the characterization of logistics management of fresh agricultural product enterprises and to conduct a reasonable performance evaluation.

In terms of research on the construction of an evaluation index system, scholars both domestically and abroad have conducted extensive research on logistics performance evaluations of fresh agricultural product enterprises (*Raut et al., 2019*; *Bai & Sarkis, 2020*; *Zhang, Li & Yao 2021*). *Tamimi, Sundarakani & Vel (2010)* defined the concept and importance of cold chain logistics and discussed its impact on other industries. *Hsiao, Chen & Chin (2017)* highlighted the need for fresh foods to be transported at low temperatures to prevent decay and reduce operating costs. *Joshi, Banwet & Shankar (2011)* developed a framework for evaluating the advantages and disadvantages of cold chain logistics operations in enterprises and identified methods for improving profitability. *Kuo & Chen (2010)* emphasized the importance of using transportation equipment and packaging appropriately, strictly controlling temperature, and improving real-time monitoring and abnormal alarm functions to prevent accidents during the circulation of fresh food. *Kashav et al. (2018)* designed a cold chain logistics performance evaluation index system to improve operational efficiency and facilitate the improvement of cold chain logistics organization modes. *Kumar et al. (2020)* further studied the evaluation methods and effects of cold chain logistics from the perspective of stability.

Regarding evaluation method research, assigning an appropriate weight to each evaluation index is a prerequisite for scientific evaluation. Two categories of weight calculation methods are commonly used: subjective and objective. Subjective weighting methods include the analytic hierarchy process (*Zha et al., 2018*; *Zeng et al., 2022*), G1 method (*Gu et al., 2021*), and Delphi algorithm (*Dawood et al., 2021*). Objective weighting methods include the entropy weight method (*Zeng, Gu & Peng, 2023*), CRITIC weight method *Yang et al. (2020)* and coefficient of variation method (*Guo, Xu & Shao, 2022*). The analytic hierarchy process is suitable for calculating the weights of multiple indicator levels. It constructs a judgment matrix based on experts' cognitive evaluations of the evaluation object and calculates the indicator weight accordingly. However, its results are highly subjective, and the consistency of the judgment matrix requires verification. The G1 method proposed by *Yajun (2007)* is a weight calculation method based on the analytic hierarchy process that is suitable for solving complex, multifactor, and large-scale evaluation problems. Compared with the analytic hierarchy process, it reduces the workload of weight calculation, does not require pairwise comparison of indicator importance, and does not need to verify the consistency of the judgment matrix. The entropy weight method calculates weight based on historical measured data for each evaluation index. The weight coefficient is determined by the magnitude of the corresponding evaluation data

change, resulting in higher reliability and accuracy than subjective weighting methods. However, the lack of guidance from business experience may distort the indicator weights, and the method is highly dependent on the samples. Compared with the CRITIC weight and coefficient of variation methods, the entropy weight method maximizes the use of attribute values of the evaluation indicators to calculate the weight coefficients of various indicators. To mitigate the bias of a single weighting method, multiple weight calculation methods are typically combined to obtain comprehensive weights (*Liu et al., 2020*; *Zhao et al., 2022*). The fusion method of comprehensive weight calculation differs based on the weight calculation principles, such as those of the weight average and linear weight methods.

Numerous scholars have contributed to the research on fresh agricultural product logistics performance evaluation, including the construction of evaluation index systems, determination of index weights, and comprehensive evaluation methods. However, existing fresh agricultural product logistics performance evaluation has the following problems:

(1) Lack of comprehensive indicators. Current research on the performance evaluation of fresh agricultural product logistics tends to focus on the performance of a certain link or aspect without establishing a comprehensive evaluation index system. This leads to insufficient comprehensiveness of the evaluation results, making it difficult to accurately reflect the overall operational status of cold chain logistics.

(2) When calculating indicator weights, there is a bias towards using a single weighting method, which makes the evaluation results highly one-sided.

(3) Lack of standardization and comparability. Owing to the lack of unified evaluation standards and indicator systems, different studies often use different evaluation indicators and methods, resulting in poor comparability of evaluation results. This makes it difficult for different enterprises or institutions to effectively compare and reference their evaluation results, limiting the further application and promotion of cold chain logistics performance evaluation.

Based on the above issues, this article proposes a comprehensive evaluation system for the sustainable development of fresh agricultural product logistics enterprises based on a combination empowerment-TOPSIS method. Based on the characteristics of fresh agricultural products, such as vulnerability and decay, a performance evaluation index system for cold chain logistics enterprises was constructed to reduce the waste of fresh agricultural products, protect the environment, and improve the market competitiveness of logistics enterprises. To solve the problem of insufficient single weighting, a combination weighting method was created to determine the weights of each indicator. This combination weighting method uses the G1 and entropy weight methods to calculate subjective and objective weights of evaluation indicators and calculates the weights of each indicator combination with the goal of minimizing the deviation of subjective and objective weighted attributes. This not only reflects the experience of experts in different indicators but also fully utilizes the information characteristics provided by the data itself, thereby obtaining indicator weight values that are more in line with actual operation. Finally, the TOPSIS technique is used to calculate the relative proximity of fresh agricultural product enterprises, providing a new concept for performance evaluation. The results show that the proposed

model can effectively evaluate the performance of fresh agricultural product logistics enterprises and provide a theoretical basis for enterprise logistics management.

The remainder of this article is organized as follows. 'Materials & Methods' introduces the index system establishment process, 'Methods' proposes the comprehensive performance evaluation model, 'Sample Analysis' explores the application of the proposed method, 'Comparative Analysis' presents a comparative analysis of evaluation results, and 'Conclusions' summarizes the practical and theoretical significance of the proposed method and its future application prospects.

# MATERIALS & METHODS

## Performance evaluation index system of fresh agricultural product logistics enterprises

When establishing a logistics performance evaluation index system, the unique characteristics of fresh agricultural products must be fully considered. First, owing to the perishability and susceptibility of fresh agricultural products to spoilage, low temperatures and high humidity must be maintained throughout the logistics process to ensure the quality and taste of the products. Second, the production and sales of fresh agricultural products have strong seasonality; therefore, the logistics supply chain must respond quickly to ensure that products reach customers in a timely manner. In addition, due to the complexity of transporting fresh agricultural products, high-level technology and equipment support are required, and logistics personnel need to be proficient in operation and management skills. Therefore, when establishing a logistics performance evaluation index system for fresh agricultural product enterprises, these characteristics and requirements must be fully considered, and the logistics performance of enterprises should be comprehensively evaluated from multiple perspectives. At the same time, to ensure a scientific and standardized evaluation system, this study referred to China's relevant cold chain logistics standards and specifications and rigorously studied and demonstrates the construction of indicators and the determination of weights. After comprehensively considering the cold supply chain capability, service quality, economic benefits, information technology level, and development capability of fresh agricultural product enterprises, a logistics performance evaluation index system for fresh agricultural product enterprises was constructed, as shown in Table 1, which includes five primary indicators ($A_1$, $A_2$, $A_3$, $A_4$, $A_5$) and 16 secondary indicators ($A_{11}$–$A_{52}$). Below are specific explanations for each indicator:

(1) Cold supply chain capability ($A_1$): The cold supply chain capability of an enterprise refers to its ability to complete the entire logistics service around its core. In this study, the utilization rate of cold storage, turnover rate of cold storage, freight loss rate of cold chain storage and transportation, and storage fresh-keeping capacity were considered as the evaluation indicators under the cold supply chain capacity dimension.

(2) Service quality ($A_2$): Various customer situations in agricultural product distribution are related to enterprise development. Only by meeting the needs of customers can their approval be gained, allowing enterprises to develop over the long term. This is

**Table 1  Indicator system for performance evaluation of fresh agricultural product enterprises.**

| Level I indicators | Level II indicators | Indicator description |
| --- | --- | --- |
| Cold supply chain capability $A_1$ | Utilization rate of cold storage $A_{11}$ | Utilization degree of cold chain logistics enterprises' cold storage |
| | Cold storage turnover rate $A_{12}$ | Reflect the circulation efficiency of fresh agricultural products in cold storage of cold chain logistics enterprises |
| | Storage fresh-keeping capacity $A_{13}$ | Fresh agricultural products need to be stored at low temperature |
| | Distribution and delivery loss rate of cold chain storage $A_{14}$ | Transportation integrity of fresh agricultural products. |
| Service quality $A_2$ | Delivery accuracy $A_{21}$ | Accuracy of agricultural product allocation and delivery |
| | Delivery timeliness rate $A_{22}$ | Timeliness of distribution of agricultural products |
| | Customer satisfaction $A_{23}$ | Customer satisfaction with the enterprise |
| | Freshness of agricultural products $A_{24}$ | It concerns the reputation of the enterprise |
| Economic performance $A_3$ | Distribution cost $A_{31}$ | Cost of transportation vehicles, personnel and refrigeration facilities |
| | Net profit growth rate $A_{32}$ | Sustainable profitability of agricultural product logistics distribution enterprises |
| | Asset current ratio $A_{33}$ | Ratio between current assets and current liabilities |
| | Return on total assets $A_{34}$ | Ratio of total remuneration before interest and tax to total average assets |
| Informatization degree $A_4$ | Information sharing degree $A_{41}$ | The relationship between the information of various departments and employees of the enterprise and the information of fresh agricultural products |
| | Timeliness of temperature information $A_{42}$ | Fresh agricultural products are perishable, so real-time temperature information should be paid attention to at any link. |
| Developing capacity $A_5$ | Equipment input rate $A_{51}$ | It reflects the importance that enterprises attach to equipment |
| | Order growth rate A52 | It reflects changes in market demand |

further refined into four secondary indicators: delivery accuracy, delivery timeliness, customer satisfaction, and freshness of agricultural products.

(3) Economic benefits ($A_3$): Economic benefits form the core of an enterprise's performance evaluation. Economic efficiency is the key to the performance evaluation of for-profit enterprises. This includes distribution cost, net profit growth rate, asset flow rate A33, and return on total assets.

(4) Informatization degree ($A_4$): In the Internet era, the degree of informatization directly affects the performance of fresh agricultural product logistics enterprises. Through the Internet, customers can become closely connected to enterprises, saving time and improving logistics efficiency. This is further subdivided into information sharing degree and temperature information timeliness.

(5) Development ability ($A_5$): Development ability relates to the development potential of an enterprise and is an important indicator for measuring the level of an enterprise's ability. It reflects the potential formed by a fresh agricultural product enterprise through

the continuous improvement of its production operations and continuous adaptation to market demand. This includes equipment input rate and order growth rate.

## METHODS

As mentioned previously, frequently used indicator weight calculation methods include both subjective and objective weights. The subjective method relies too much on expert experience, while the objective method relies heavily on samples, lacks guidance from business experience, and the weight may be distorted, resulting in invalid results. Therefore, single weighting methods exhibit strong subjectivity or objectivity. This study improved the reliability of the evaluation results by calculating the weights of an evaluation index combination and used the TOPSIS method to comprehensively evaluate the performance of cold chain logistics enterprises. The specific evaluation process includes four steps: (1) standardize the original data of the evaluation index with the range method; (2) use the G1 and entropy weight methods to calculate the subjective and objective weights of evaluation indicators, respectively; (3) calculate the combination weight of the evaluation indicators; and (4) construct the weighted judgment matrix and use the TOPSIS method to comprehensively evaluate the evaluation object.

### Subjective weighting method–G1

The G1 method is a simple and effective subjective weighting method that does not require consistency testing. The G1 method is a more perfect weight calculation method based on the analytic hierarchy process (AHP), avoiding the disadvantage that the judgment matrix of AHP cannot pass the consistency test because of the large number of evaluation indicators. The detailed calculation steps are as follows:

Step 1: Establish the order relationship. Assume that the evaluation indicator set { c1, c2, $\cdots$, cn } contains $n$ indicators at the same level in the indicator system, and $n \geq 2$. In combination with experts' opinions, determine the index sequence as follows:

(1) Experts select the most important evaluation index from the evaluation index set { c1, c2, $\cdots$, cn } and record it as $c_1^*$;

(2) Select the next most important evaluation indicator from the $n$-1 remaining evaluation indicators in the evaluation indicator set and record it as $c_2^*$, after $n$-1 selections. The last evaluation index is marked as $c_n^*$;

(3) Thus, the order relation of the evaluation index set { c1, c2, $\cdots$, cn } can be determined as follows:

$$c_1^* \geq c_2^* \geq \cdots \geq c_{n-1}^* \geq c_n^* \tag{1}$$

The reconstituted set { $c_1^*$, $c_2^*$, $\cdots$, $c_{n-1}^*$, $c_n^*$ } is called the evaluation indicator set after determining the order relationship, and then the importance ranking between adjacent indicators can be obtained.

Step 2: Quantitative analysis of the importance of adjacent indicators quantifies the importance of adjacent evaluation indicators according to Table 2, which can be expressed as follows:

$$r_k = \frac{\alpha_{k-1}^*}{\alpha_k^*} \tag{2}$$

**Table 2 Assignment reference.**

| $r_k$ | Description of assignment |
|---|---|
| 1.0 | $c_{k-1}^*$ and $c_k^*$ are equally important |
| 1.2 | $c_{k-1}^*$ is slightly more important than $c_k^*$ |
| 1.4 | $c_{k-1}^*$ is obviously more important than $c_k^*$ |
| 1.6 | $c_{k-1}^*$ is more important than $c_k^*$ |
| 1.8 | $c_{k-1}^*$ is more extraordinary important than $c_k^*$ |
| 1.1, 1.3, 1.5, 1.7 | The median value of the above two adjacent judgments |

where $r_k$ represents the relative importance ratio between adjacent evaluation indicators $c_{k-1}^*$ and $c_k^*$; the value range of $k$ is $[2, n]$; and $\alpha_{k-1}^*$ and $\alpha_k^*$ represent the weights of adjacent evaluation indicators $c_{k-1}^*$ and $c_k^*$, respectively. According to common cultural terms, the $r_k$ assignment based on the 9-level mood operator is established (*Yajun, 2007*), as shown in Table 2.

Step 3: Calculate the index weight. According to the given assignment of $r_k$, the weight calculation formula for the evaluation index $c_n^*$ is

$$\alpha_n^* = \left(1 + \sum_{k=2}^{n} \prod_{i=k}^{n} r_i\right)^{-1} \tag{3}$$

$$\alpha_{k-1}^* = r_k \alpha_k^* \quad (k = n, n-1, \ldots, 3, 2) \tag{4}$$

where $\alpha_n^*$ is the weight of $c_n^*$. Eq. (3) gives the subjective weight value obtained based on expert decisions, and Eq. (4) gives the subjective weight of each index.

Thus, we obtain the subjective weight vector $\alpha = (\alpha_1, \alpha_{2}, \ldots, \alpha_n)$.

## Objective weighting method: entropy weighting method

Shannon first introduced entropy into information theory to measure system uncertainty. Information entropy quantitatively describes the amount of information contained in a piece of information. The entropy weight method determines the weight based on the information entropy of the evaluation index (*Cui & Ye, 2018*). In general, the smaller the information entropy of the evaluation index, the greater the variation degree of the index value, the more information is provided, and the greater its weight in the comprehensive evaluation. On the contrary, a larger information entropy of an indicator indicates that the variation degree of the indicator value is smaller, the amount of information provided is less, and its weight in the comprehensive evaluation is smaller. It can be seen that the entropy weight method is an objective evaluation method based on the actual data of evaluation indicators, and its evaluation results are more objective. The calculation steps are as follows:

Step 1: Raw data preprocessing. According to the evaluation indicators and raw data of the evaluation object, the initial evaluation matrix is established as follows:

$$X = \left(x_{ij}\right)_{m \times n} = \begin{pmatrix} x_{11} & x_{12} & \cdots & x_{1n} \\ x_{21} & x_{22} & \cdots & x_{2n} \\ \vdots & \vdots & \ddots & \vdots \\ x_{m1} & x_{m2} & \cdots & x_{mn} \end{pmatrix} \tag{5}$$

where $X$ represents the initial evaluation matrix, $x_{ij}$ represents the raw data of the $j$th index of the $i$th evaluation object, whose value is obtained from the operation statistics of the logistics enterprises in a period of time, and $m$ and $n$ represent the numbers of evaluation objects and indicators, respectively.

The original data obtained are standardized; that is, the range method is used to conduct dimensionless and isotropic processing for each datum. To avoid a situation where the dimensionless data are equal to zero, 0.0001 is added to the entire formula for translation processing.

For positive indicators, also known as "benefit-oriented" indicators, the standardized equation is

$$y_{ij} = \frac{x_{ij} - \min_{1 \le i \le m}\{x_{ij}\}}{\max_{1 \le i \le m}\{x_{ij}\} - \min_{1 \le i \le m}\{x_{ij}\}} + 0.0001 \tag{6}$$

For negative indicators, the standardized formula is

$$y_{ij} = \frac{\max_{1 \le i \le m}\{x_{ij}\} - x_{ij}}{\max_{1 \le i \le m}\{x_{ij}\} - \min_{1 \le i \le m}\{x_{ij}\}} + 0.0001 \tag{7}$$

where $y_{ij}$ represents the standardized evaluation index data, and the logistics performance standardization matrix can be obtained as follows:

$$Y = \begin{pmatrix} y_{11} & y_{12} & \cdots & y_{1n} \\ y_{21} & y_{22} & \cdots & y_{2n} \\ \vdots & \vdots & \ddots & \vdots \\ y_{m1} & y_{m2} & \cdots & y_{mn} \end{pmatrix} \tag{8}$$

Step 2: Calculate the proportion of the $i$th evaluation object:

$$p_{ij} = \frac{y_{ij}}{\sum_{i=1}^{m} y_{ij}} \tag{9}$$

Step 3: Calculate information entropy:

$$e_j = -\frac{1}{\ln(m)} \sum_{i=1}^{m} p_{ij} \ln\left(p_{ij}\right) \tag{10}$$

Step 4: Calculating the entropy weight:

$$\beta_j = \frac{1 - e_j}{n - \sum_{i=1}^{m} e_j} \tag{11}$$

where $\beta_j$ represents the entropy weight of the $j$th evaluation indicator. Then, the objective weight vector $\beta$ on the index set can be obtained as follows:

$$\begin{cases} \beta = (\beta_1, \beta_2, \cdots, \beta_n) \\ 0 \leq \beta_j \leq 1, \sum_{j=1}^{n} \beta_j = 1 \end{cases} \tag{12}$$

## Combined weighting

It can be seen from above section that the subjective weight value is calculated as $\alpha = (\alpha_1, \alpha_2, \cdots, \alpha_n)$, and the objective weight value is calculated as $\beta = (\beta_1, \beta_2, \cdots, \beta_n)$. Assume the coefficients in the combination weight are defined as t and $\tau$, respectively, and $t \times \tau \geq 0$, $t + \tau = 1$. Then, the combination weight $w_j$ is calculated as

$$w_j = t\alpha_j + \tau\beta_j \, (j = 1, 2, \ldots, n) \tag{13}$$

According to Eq. (3) the combined weighting attribute values of any evaluation object can be obtained. Therefore, the subjective weighting attribute of the $j$th index of the $i$th evaluation object can be expressed as $t\alpha_j y_{ij}$, and the objective weighting attribute can be expressed as $\tau\beta_j y_{ij}$. Thus, the degree of deviation can be expressed as

$$Z_i = \sum_{j=1}^{n} (t\alpha_j y_{ij} - \tau\beta_j y_{ij})^2 \, (i = 1, 2, \ldots, m) \tag{14}$$

where $Z_i$ represents deviation from the $i$th evaluated object. Clearly, the smaller the deviation degree, the more consistent the two types of weight attributes. We establish a weight combination optimization model by minimizing the deviation of the subjective and objective weighted attributes as the objective function.

$$\begin{cases} minZ_i = \sum_{i=1}^{m} Z_i = \sum_{j=1}^{n} (t\alpha_j y_{ij} - \tau\beta_j y_{ij})^2 \, (i = 1, 2, \ldots, m) \\ s.t. \begin{cases} t + \tau = 1, t \times \tau \geq 0 \\ 0 \leq t \leq 1 \\ 0 \leq \tau \leq 1 \end{cases} \end{cases} \tag{15}$$

The solution process of this optimization model is given below.
Build Lagrange function:

$$L(t, \tau, \mu) = \sum_{i=1}^{m} \sum_{j=1}^{n} (t\alpha_j y_{ij} - \tau\beta_j y_{ij})^2 + \mu(t + \tau - 1) \tag{16}$$

where $\mu$ represents the Lagrange multiplier. Then,

$$\frac{\partial L(t, \tau, \mu)}{\partial t} = 2\sum_{i=1}^{m} \sum_{j=1}^{n} t\alpha_j^2 y_{ij}^2 - 2\sum_{i=1}^{m} \sum_{j=1}^{n} \left(\tau\alpha_j\beta_j y_{ij}^2\right) + \mu \tag{17}$$

$$\frac{\partial L(t, \tau, \mu)}{\partial \tau} = 2\sum_{i=1}^{m} \sum_{j=1}^{n} \tau\beta_j^2 y_{ij}^2 - 2\sum_{i=1}^{m} \sum_{j=1}^{n} \left(t\alpha_j\beta_j y_{ij}^2\right) + \mu \tag{18}$$

$$\frac{\partial L(t,\tau,\mu)}{\partial \mu} = t + \tau - 1 \tag{19}$$

Letting Eqs. (17)–(19) be equal to zero, the simultaneous equation can be solved to obtain the weight coefficients $t$ and $\tau$ as follows:

$$t = \frac{\sum_{i=1}^{m}\sum_{j=1}^{n}\beta_j y_{ij}^2(\alpha_j + \beta_j)}{\sum_{i=1}^{m}\sum_{j=1}^{n}y_{ij}^2(\alpha_j + \beta_j)^2} \tag{20}$$

$$\tau = \frac{\sum_{i=1}^{m}\sum_{j=1}^{n}\alpha_j y_{ij}^2(\alpha_j + \beta_j)}{\sum_{i=1}^{m}\sum_{j=1}^{n}y_{ij}^2(\alpha_j + \beta_j)^2} \tag{21}$$

Solve Eqs. (20) and (21) to obtain the coefficients $t$ and $\tau$, and substitute $\tau$ and $t$ into Eq. (13) to calculate the weight value. The proposed unified combination weighting method solves the problem of average distribution, which results in an unreasonable evaluation index weight.

## TOPSIS evaluation method

TOPSIS, *i.e.,* "the distance between good and bad solutions", is suitable for comparative analysis of multiple evaluation objects. The basic process of the algorithm is to build a weighted judgment matrix based on the standardized original data matrix, calculate the positive and negative ideal solutions in the evaluation indicators, and then calculate the Euclidean distance between each evaluation object and the positive and negative ideal solutions, which forms the basis for the evaluation. The TOPSIS method is relatively flexible in terms of data requirements and can adapt better to the availability and applicability of data. Therefore, based on the characteristics of the research data, we selected the TOPSIS method for comprehensive evaluation of fresh logistics enterprises. However, for the sample points on the bisector of "positive and negative ideal solutions", the classical TOPSIS method cannot distinguish between good and bad. *Niu et al. (2021)* improved TOPSIS by introducing "virtual negative ideal solutions" instead of "negative ideal solutions" to make the evaluation results more reasonable. The steps of the algorithm are as follows:

Step 1: Build a weighted judgment matrix

$$S = YW = \begin{pmatrix} y_{11} & y_{12} & \cdots & y_{1n} \\ y_{21} & y_{22} & \cdots & y_{2n} \\ \vdots & \vdots & \ddots & \vdots \\ y_{m1} & y_{m2} & \cdots & y_{mn} \end{pmatrix} \begin{pmatrix} w_1 & 0 & \cdots & 0 \\ 0 & w_2 & \cdots & 0 \\ \vdots & \vdots & \ddots & \vdots \\ 0 & 0 & \cdots & w_n \end{pmatrix} = \begin{pmatrix} s_{11} & s_{12} & \cdots & s_{1n} \\ s_{21} & s_{22} & \cdots & s_{2n} \\ \vdots & \vdots & \ddots & \vdots \\ s_{m1} & s_{m2} & \cdots & s_{mn} \end{pmatrix} \tag{22}$$

where $S$ represents the weighted judgment matrix, $W$ represents the evaluation index weight matrix, $w_j$ represents the combined weight of each index, and $s_{ij}$ represents the $j$th index weight value of the $i$th evaluation object.

Step 2: Determine the positive, negative, and virtual negative ideal solutions $S^+$, $S^-$, $S^*$:

$$S^+ = \max_{1 \le i \le m} s_{ij} = \left(s_1^+, s_2^+, \ldots, s_n^+\right) \tag{23}$$

$$S^- = \min_{1 \le i \le m} s_{ij} = \left(s_1^-, s_2^-, \ldots, s_n^-\right) \tag{24}$$

$$S^* = 2S^- - S^+ = \left(2s_1^- - s_1^+, 2s_2^- - s_2^+, \ldots, 2s_n^- - s_n^+\right) = \left(s_1^*, s_2^*, \ldots, s_n^*\right) \tag{25}$$

where the value range of $j$ is from 1 to $n$, $S^*$ represents "virtual negative ideal solution", and $s_j^+$ and $s_j^-$ represent the positive and negative ideal solutions of the $j$th evaluation indicator, respectively.

Step 3: Calculate the Euclidean distance between the object to be evaluated and the ideal solution.

Euclidean distance between the $i$th evaluation object and $S^+$:

$$D_i^+ = \sqrt{\sum_{j=1}^{n} \left(s_j^+ - s_{ij}\right)^2} \quad i = 1, 2, \ldots, m. \tag{26}$$

Euclidean distance between the $i$th evaluation object and $S^*$:

$$D_i^* = \sqrt{\sum_{j=1}^{n} \left(s_j^* - s_{ij}\right)^2} \quad i = 1, 2, \ldots, m \tag{27}$$

where $D_i^+$ and $D_i^*$ represent the Euclidean distance between the $i$th object to be evaluated and $S^+$ and $S^*$, respectively.

Step 4: Calculating the relative proximity:

$$C_i = \frac{D_i^*}{D_i^* + D_i^+} \tag{28}$$

where $C_i$ denotes the relative proximity of the $i$th object to be evaluated. Obviously, $0 \le C_i \le 1$, and the larger $C_i$ is, the smaller $D_i^+$ is, and the better the evaluation object is.

## SAMPLE ANALYSIS

For product transportation in the sales process of a fresh agricultural product production base, the comprehensive capacity of five fresh agricultural product cold chain logistics enterprises (A, B, C, D, E, and F) was evaluated, and the enterprise with the strongest comprehensive capacity was selected for cooperation. The index evaluation value for each logistics enterprise was obtained using an expert scoring method and enterprise operation data. Further details are presented in Table 3.

### Index weight calculation

Experts in the field of fresh agricultural product logistics were organized to calculate the subjective weight using Eqs. (1)~(4), and the objective weight was obtained using Eqs. (5)~(12). Equations (13)~(14) are used to calculate the combination weight coefficients $t$ and $\tau$, and we can get $t = 0.3642$, $\tau = 0.6358$. Furthermore, we obtained

**Table 3  Original data, positivity and negativity of evaluation indicators.**

| Indicator | Indicator attribute | Raw data | | | | | |
|---|---|---|---|---|---|---|---|
| | | A | B | C | D | E | F |
| $A_{11}$ | + | 0.96 | 0.75 | 0.72 | 0.83 | 0.61 | 0.76 |
| $A_{12}$ | + | 0.92 | 0.86 | 0.75 | 0.89 | 0.82 | 0.87 |
| $A_{13}$ | + | 0.93 | 0.81 | 0.76 | 0.87 | 0.81 | 0.72 |
| $A_{14}$ | - | 0.65 | 0.81 | 0.74 | 0.71 | 0.83 | 0.86 |
| $A_{21}$ | + | 0.92 | 0.83 | 0.75 | 0.89 | 0.84 | 0.81 |
| $A_{22}$ | + | 0.96 | 0.81 | 0.72 | 0.90 | 0.83 | 0.82 |
| $A_{23}$ | + | 9 | 8 | 7 | 8 | 7 | 7 |
| $A_{24}$ | + | 9 | 8 | 8 | 8 | 7 | 8 |
| $A_{31}$ | - | 62 | 92 | 85 | 66 | 65 | 86 |
| $A_{32}$ | + | 0.95 | 0.84 | 0.67 | 0.92 | 0.87 | 0.85 |
| $A_{33}$ | + | 0.96 | 0.83 | 0.76 | 0.90 | 0.81 | 0.83 |
| $A_{34}$ | + | 0.87 | 0.81 | 0.75 | 0.87 | 0.82 | 0.81 |
| $A_{41}$ | + | 0.90 | 0.85 | 0.78 | 0.86 | 0.86 | 0.82 |
| $A_{42}$ | + | 0.93 | 0.87 | 0.71 | 0.90 | 0.80 | 0.84 |
| $A_{51}$ | + | 0.92 | 0.86 | 0.79 | 0.87 | 0.81 | 0.63 |
| $A_{52}$ | + | 0.91 | 0.83 | 0.81 | 0.88 | 0.82 | 0.86 |

the combined weights of the evaluation indices. Table 4 shows the values of three types of weights, where $\alpha_j$ represents the subjective weight of logistics enterprise performance evaluation indicators calculated using the G1 method, $\beta_j$ represents the objective weight calculated using the entropy weight method, and $w_j$ represents the weight calculated using our combination weighting method.

From Table 4, it can be observed that the subjective and objective weights calculated based on the G1 and entropy weight methods have significant differences for certain indicators. For example, for the indicator Delivery accuracy $A_{21}$, the subjective and objective weights are 0.019 and 0.132, respectively. In the daily operations of logistics enterprises, the importance of this indicator should be higher than the Utilization rate of cold storage $A_{11}$ and Cold storage turnover rate $A_{12}$. Therefore, it can be seen that the G1 method lacks practical operational experience in calculating indicator weights. Fusing the weights obtained from the two methods not only fully utilizes the information characteristics provided by the data itself but also reflects the experience of experts on different indicators, obtaining indicator weight values that are more in line with actual operation.

In the daily operation of logistics enterprises, the higher the order demand, the higher the customer satisfaction with the enterprise and the better the economic benefits of the enterprise. Therefore, order growth rate is one of the core indicators of concern in the performance evaluation of various logistics enterprises. Table 4 shows that in the performance evaluation index system for fresh agricultural product logistics enterprises constructed in this study, the weight of order growth rate accounts for the largest

**Table 4  Subjective weights, objective weights and combined weights.**

| Index | Subjective weight $\alpha_j$ | Objective weight $\beta_j$ | Combination weight $w_j$ |
|---|---|---|---|
| A11 | 0.272 | 0.057 | 0.135 |
| A12 | 0.081 | 0.057 | 0.065 |
| A13 | 0.087 | 0.054 | 0.066 |
| A14 | 0.140 | 0.088 | 0.107 |
| A21 | 0.019 | 0.132 | 0.090 |
| A22 | 0.021 | 0.163 | 0.111 |
| A23 | 0.059 | 0.131 | 0.104 |
| A24 | 0.029 | 0.074 | 0.056 |
| A31 | 0.262 | 0.062 | 0.135 |
| A32 | 0.326 | 0.146 | 0.211 |
| A33 | 0.192 | 0.126 | 0.150 |
| A34 | 0.426 | 0.162 | 0.258 |
| A41 | 0.042 | 0.169 | 0.123 |
| A42 | 0.036 | 0.428 | 0.285 |
| A51 | 0.162 | 0.226 | 0.202 |
| A52 | 0.159 | 0.372 | 0.294 |

**Table 5  Performance evaluation structure of logistics enterprises.**

| Evaluation object | $D_i^+$ | $D_i^*$ | $C_i$ | Rank |
|---|---|---|---|---|
| A | 0.0926 | 0.5621 | 0.8629 | 1 |
| B | 0.1192 | 0.5391 | 0.8179 | 3 |
| C | 0.1362 | 0.5326 | 0.8035 | 5 |
| D | 0.1156 | 0.5476 | 0.8267 | 2 |
| E | 0.1206 | 0.5368 | 0.8152 | 4 |
| F | 0.1569 | 0.4872 | 0.7560 | 6 |

proportion, which is consistent with the current operational needs of logistics enterprises. This further demonstrates the effectiveness of the proposed combined weight model.

## Comprehensive performance evaluation

Based on the raw data in Table 3 and the combined weights of the evaluation index in Table 4, we can calculate the Euclidean distance and relative proximity between each evaluation object and the ideal value using Eqs. (21)–(28). Furthermore, by calculating relative proximity, we can determine the performance rankings of cold chain logistics enterprises. The results are shown in Table 5, where $D_i^+$ represents the Euclidean distance between each logistics enterprise to be evaluated and the virtual positive ideal solution, $D_i^*$ represents the Euclidean distance from the virtual negative ideal solution, and $C_i$ represents the relative proximity value. The higher the $C_i$ value, the better the object to be evaluated. According to the $C_i$ values in Table 5, the cold chain logistics enterprises of fresh agricultural products rank as follows: $A \succ D \succ B \succ E \succ C \succ F$. Therefore, the agricultural product production base should choose logistics enterprise A for cooperation.

**Table 6  Performance evaluation structure of logistics enterprises with different weight methods.**

|  | A | B | C | D | E | F |
|---|---|---|---|---|---|---|
| $\alpha_i$(Subjective weight) | 1.0000 | 0.6838 | 0.5909 | 0.8749 | 0.6983 | 0.6778 |
| $\beta_i$(Objective weight) | 1.0000 | 0.7870 | 0.5655 | 0.8925 | 0.7093 | 0.6816 |

## COMPARATIVE ANALYSIS

Several comparative analyses were conducted to further demonstrate the feasibility and effectiveness of the proposed method. First, compared to using the integrated weight system, we only considered the subjective and objective weights in the evaluation process separately. The results are shown in Table 6, where $\alpha_i$ represents the subjective weight value, $\beta_i$ represents the objective weight value, and $A$, $B$, $C$, $D$, $E$, and $F$ represent six different logistics enterprises.

As can be seen from the evaluation results, the ranking under subjective weight is $A \succ D \succ E \succ B \succ F \succ C$, and that under objective weight is $A \succ D \succ B \succ E \succ F \succ C$. There are slight deviations between the ranking results of the two weighting methods and those of the proposed method. Compared to simply considering the subjective or objective weight, the combined weight calculation method can more effectively integrate expert opinions and information of the data itself, making the weight given to the results more reasonable.

To verify the effectiveness of this method, we compared it with the classical TOPSIS and VIKOR methods. Similarly, based on relative proximity, the rankings of the five evaluation methods are presented in Table 7.

There are slight differences among the five evaluation methods, but the top-2 rankings are always A and D, and enterprise A is always the best scheme, which verifies the scientificness and effectiveness of the G1-Entropy-TOPSIS method proposed in this article. At the same time, compared with the classical TOPSIS method, this study addresses the problem that the sample points on the vertical line of the positive and negative ideal solutions cannot be distinguished by introducing a virtual negative ideal solution instead of a negative ideal solution. Compared to the TOPSIS method, the VIKOR method has one more decision-making mechanism coefficient, which makes it more subjective. In addition, there may be more than one optimal solution after ranking using the VIKOR method, whereas TOPSIS provides only one optimal solution. Figure 1 shows a comparison of the closeness under several schemes, where different colored lines represent different schemes, and the red line represents the proposed method. Lines of the same color create a closed loop, and the larger the area enclosed in this loop, the better the solution. These results indicate that our solution is generally the best.

This study has significant practical and theoretical implications for the field of cold chain logistics, particularly in the context of fresh agricultural products. The combination of the G1 method, entropy weight method, and TOPSIS technique not only offers a practical solution for the evaluation of cold chain logistics enterprises but also contributes to the enrichment of evaluation methodologies in logistics and supply chain management. The theoretical implications can be summarized as follows:

**Table 7  Comparison of results.**

| Application method | Ranking results |
| --- | --- |
| G1-Entropy-TOPSIS | $A \succ D \succ B \succ E \succ C \succ F$ |
| G1-TOPSIS | $A \succ D \succ E \succ B \succ F \succ C$ |
| Entropy-TOPSIS | $A \succ D \succ B \succ E \succ F \succ C$ |
| Classical TOPSIS method | $A \succ D \succ B \succ F \succ C \succ E$ |
| VIKOR method (*Opricovic & Tzeng, 2004*) | $A \succ D \succ B \succ E \succ F \succ C$ |

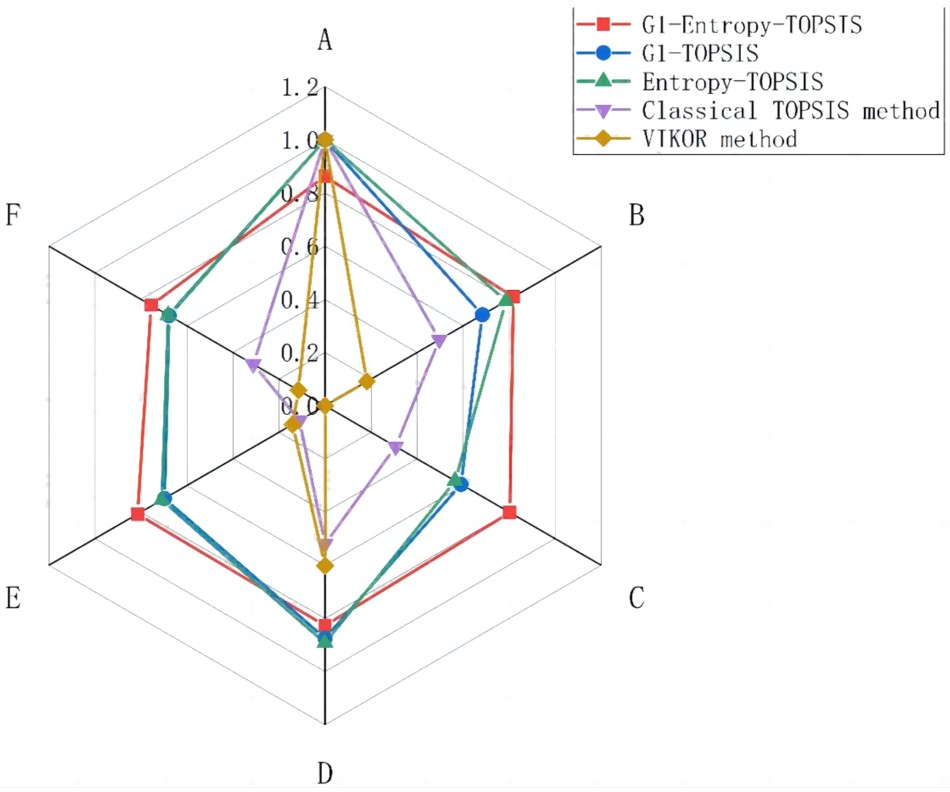

**Figure 1  Comparative analysis chart of closeness.**

(1) Enrichment of TOPSIS methodology: This research enriches the application of the TOPSIS method by introducing a novel approach to combine subjective and objective weightings. Traditional TOPSIS methods often rely solely on either subjective or objective weights, which may introduce biases or result in fluctuating outcomes owing to data variations. Our approach minimizes these issues by using both types of weights, thereby enhancing the theoretical foundation of the TOPSIS technique.

(2) Multidimensional evaluation model: The construction of a comprehensive evaluation system encompassing five key dimensions–cold supply chain capacity, service quality, economic efficiency, degree of informatization, and development ability–presents a multidimensional framework for assessing the performance of cold chain logistics

enterprises. This extends our understanding of logistics performance evaluation by considering a broader set of factors relevant to the fresh agricultural product industry.

(3) Reduction of subjectivity: The integration of the G1 method, an improvement over the traditional AHP, helps reduce the subjectivity of the weight assignment in the evaluation process. This theoretical advancement enhances the objectivity and reliability of logistics performance assessments, rendering them applicable to a wider range of scenarios.
The practical implications can be summarized as follows:

(1) Guidance for logistics enterprises: The performance evaluation index system and comprehensive evaluation model provide valuable guidance for cold chain logistics enterprises. By assessing their performance across various dimensions, logistics companies can identify their strengths and weaknesses, enabling them to make informed decisions regarding future development directions and investment priorities.

(2) Market competitiveness enhancement: As the fresh agricultural product industry continues to grow, market competitiveness has become paramount. This research offers a practical tool for logistics enterprises to enhance their competitiveness by evaluating and improving their cold chain logistics capabilities, service quality, and economic efficiency. This can help to align them with consumers' evolving needs.

(3) Environmental and resource benefits: The implementation of effective cold chain logistics practices based on this evaluation model can contribute to the reduction of waste in the transportation and distribution of fresh agricultural products. This, in turn, supports environmental protection, minimizes resource waste, and addresses critical concerns in the industry.

(4) All-round development promotion: The evaluation system not only benefits logistics enterprises but also promotes the overall development of the cold chain logistics industry. By raising technical standards, improving efficiency, and enhancing reputation, this research facilitates the growth of the entire industry and its capacity to meet the demands of a rapidly evolving market.

## CONCLUSIONS

From the perspective of reducing the waste of fresh agricultural products, protecting the environment, and improving the competitiveness of logistics enterprises in the market, this study constructed a performance evaluation index system for cold chain logistics enterprises and established a comprehensive evaluation model. This method uses the G1 and entropy weight methods to calculate the subjective and objective weights of the evaluation indicators, respectively, and determines the combined weight by minimizing the deviation of the subjective and objective weighted attributes. The combination of these two methods not only reduces the subjectivity of the G1 method but also reduces the fluctuation of the weight value caused by data changes.

In terms of theoretical significance, this study expanded the TOPSIS method, combining it with G1 and entropy weight methods, enriching the application fields of the TOPSIS method, and presenting important theoretical significance. In terms of practical significance, on the one hand, the evaluation index system can evaluate the comprehensive capacity of cold chain logistics enterprises, providing a reference for their own choices and

future development direction; on the other hand, it can promote logistics enterprises and their ability, technical level, and reputation in a comprehensive manner. However, owing to limited resources, our work has certain limitations, which are detailed as follows:

(1) Insufficient data acquisition and processing. The logistics performance evaluation of fresh agricultural products requires a large amount of data support, including logistics transportation, temperature, and time data. However, current research faces difficulties in obtaining data, and there are certain limitations in data processing and analysis methods that cannot fully tap into the potential of data.

(2) Insufficiently comprehensive selection of performance evaluation indicators. In terms of selecting performance evaluation indicators, the main focus is on the relevant indicators of fresh agricultural product logistics enterprises themselves, without selecting indicators from their external environment, which limits the evaluation results.

With the development of artificial intelligence technology and the gradual disclosure of relevant logistics enterprise data in China, in our future work, we expect to use deep learning and big data mining technology to process data in real time to dynamically evaluate the performance of fresh agricultural product logistics enterprises.

### Funding
This work was supported by the Zhejiang Provincial Philosophy and Social Sciences Planning Project (21NDJC021Z) and the Natural Science Foundation of Ningbo city (202003N4072). The funders had no role in study design, data collection and analysis, decision to publish, or preparation of the manuscript.

### Grant Disclosures
The following grant information was disclosed by the authors:
Zhejiang Provincial Philosophy and Social Sciences Planning Project: 21NDJC021Z.
Natural Science Foundation of Ningbo city: 202003N4072.

### Competing Interests
The authors declare there are no competing interests.

### Author Contributions
- Dechao Sun conceived and designed the experiments, authored or reviewed drafts of the article, and approved the final draft.
- Xuefang Hu performed the experiments, prepared figures and/or tables, and approved the final draft.
- Bangquan Liu analyzed the data, performed the computation work, authored or reviewed drafts of the article, and approved the final draft.

### Data Availability
The raw measurements and code are available in the Supplemental Files.

## Supplemental Information

Supplemental information for this article can be found online at http://dx.doi.org/10.7717/peerj-cs.1719#supplemental-information.

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
