# Peer review of "Comprehensive evaluation for the sustainable development of fresh agricultural products logistics enterprises based on combination empowerment-TOPSIS method"

_PeerJ Computer Science, doi:10.7717/peerj-cs.1719_

## Round 0.1 · original submission · Major Revisions

Dear authors,

Your paper has been reviewed by two reviewers who asked for revisions of the paper. Please revise the paper according to comments by reviewers, mark all changes in the new version of the paper, and provide a cover letter with replies to them point to point.

**Language Note:** PeerJ staff have identified that the English language needs to be improved. When you prepare your next revision, please either (i) have a colleague who is proficient in English and familiar with the subject matter review your manuscript, or (ii) contact a professional editing service to review your manuscript. PeerJ can provide language editing services - you can contact us at copyediting@peerj.com for pricing (be sure to provide your manuscript number and title). – PeerJ Staff

·

Basic reporting

The English language in the paper is fine.
The authors must explain in more detail the goals of the research in the introduction, so it is also necessary to state which gaps this research solves. The structure of tables and figures is good. However, the explanation of them is not good, what is stated in the tables should be explained in more detail. The results must be explained in more detail. The steps of the method must be explained and the method of calculating the weights and ranking must be explained in more detail.

Experimental design

The objectives and research question must be explained in more detail in the introduction. The research is good, however, it is necessary to better explain the steps of these methods in the example because only the results and the way in which these results were obtained are listed. When theoretically explaining methods, references must be given. Thus, with the G1 method, it is not stated who created the method.

Validity of the findings

This paper contains enough contributions, but it is necessary to explain it in more detail. The listed results are fine. In conclusion, the limits of the research and guidelines for future research are missing.

Additional comments

Authors must work to explain in detail the results obtained. Then it is necessary to write a discussion where these results will be explained why they are like this and compare it with previous research.

Reviewer 2 ·

Basic reporting

First of all, I appreciate the opportunity to review the paper Comprehensive Evaluation for the sustainable development of fresh agricultural products logistics enterprises based on combination empowerment TOPSIS method. The paper deals with a very interesting problem.
Suggestions are below:


• It is necessary to understand the purpose and aim of the paper as well as its "position" in relation to previous research (also gap analysis).
• The most important results/findings must be emphasized in the abstract.
• Keywords are missing.
• The last paragraph in the introduction section should be a structure of the paper (several sentences for each section).
• Figures and tables should be better displayed and organized (font size, font style, etc).

Experimental design

• The reasons/argumentation for using the TOPSIS method must be stronger and more convincing. Why other methods were not used?
• Sections “Sample analysis” and “Comparative analysis” are not well written. A lot of details are missing.
• The separate section on Practical and theoretical implications is missing.

Validity of the findings

• Conclusion section is not on a satisfactory level. The conclusion in scientific papers is very important.
• Limitations of your research must be emphasized.
• Future research directions are missing (one sentence is not enough)
• Scientific contributions are missing.

Additional comments

.

---

## Round 0.2 · accepted · Accept

Dear authors,

Your revised version of the paper has been reviewed, and both reviewers have accepted it.

·

Basic reporting

The paper has been corrected in accordance with the review.

Experimental design

The paper has been corrected in accordance with the review.

Validity of the findings

The paper has been corrected in accordance with the review.

Additional comments

The paper has been corrected in accordance with the review.

Reviewer 2 ·

Basic reporting

The paper should be accepted for publication.

Experimental design

The paper should be accepted for publication.

Validity of the findings

The paper should be accepted for publication.

Additional comments

The paper should be accepted for publication.